# Effect of Carbon Nanotube Addition on the Interfacial Adhesion between Graphene and Epoxy: A Molecular Dynamics Simulation

**DOI:** 10.3390/polym11010121

**Published:** 2019-01-11

**Authors:** Shuangqing Sun, Shenghui Chen, Xuanzhou Weng, Fei Shan, Songqing Hu

**Affiliations:** 1School of Materials Science and Engineering, China University of Petroleum (East China), Qingdao 266580, China; sunshuangqing@upc.edu.cn (S.S.); xzweng2010@126.com (X.W.); shanfeik@163.com (F.S.); 2Institute of Advanced Materials, China University of Petroleum (East China), Qingdao 266580, China; 3School of Physics and Optoelectronic Engineering, Ludong University, Yantai 264025, China

**Keywords:** graphene, carbon nanotube, composites, pullout, molecular dynamics simulation

## Abstract

The pullout process of graphene from an epoxy/graphene composite filled with a carbon nanotube (CNT) was simulated by molecular dynamics simulations. The interaction energy and the interfacial adhesion energy were calculated to analyze the effect of CNT addition on the interfacial adhesion between the graphene and the epoxy matrix, with varying CNT radii, distances between the CNT and the graphene sheet, CNT axial directions, and the number of CNT walls. Generally, the addition of a CNT strengthens the interfacial adhesion between the graphene and the polymer matrix. Firstly, a larger CNT radius induces a stronger interfacial adhesion of graphene with the matrix. Secondly, when the CNT is farther away from the graphene sheet, the interfacial adhesion of graphene with the matrix becomes weaker. Thirdly, the CNT axial direction has little effect on the interfacial adhesion of graphene in the equilibrium structure. However, it plays an important role in the graphene pullout process. Finally, compared with a single-walled CNT, the interfacial adhesion between graphene and the matrix is stronger when a double-walled CNT is added to the matrix.

## 1. Introduction

In recent years, carbon nanotubes (CNT) and graphene have attracted significant attention due to their excellent mechanical, electrical, and thermal properties. They both show great potential for applications as nanofillers in polymer composites due to their superior properties. Composites filled with a CNT or graphene present various excellent properties [1]. In addition, a remarkable synergetic effect between a CNT and graphene in improving composite properties has been demonstrated [2,3]. 

Sumfleth et al. [4] and Yang et al. [5] detected the mechanical and thermal properties of CNT/graphene/epoxy composites. They found that CNTs and graphene have a significant synergistic effect. Kim et al. [6] found that the CNT/graphene oxide composite system has good dispersibility. Zhang et al. [7] reported that a CNT and graphene could form a more stable dispersion system when the diameter of the CNT is greater than 8 nm. Li et al. [8] found a synergistic effect of hybrid carbon nanotube–graphene oxide as a nanofiller in enhancing the mechanical properties of PVA composites. It is believed that two reasons contribute to the synergetic effect between the CNT and graphene—improved dispersion of the CNT and graphene in the matrix, and enhanced interfacial adhesion between the carbon nanoparticles and the matrix [9,10,11,12,13].

In this work, we focused on the interfacial adhesion between the carbon nanoparticles and the matrix. Many experiments have indicated that interfacial adhesion plays an important role in the synergetic effect between the CNT and graphene. Sa et al. [11] prepared a reduced graphene oxide (rGO)/CNTs/poly(methyl methacrylate) (PMMA) by the solution mixing method. Raman spectroscopy was used to study the interaction between the polymer matrix and the nanofiller. Their results showed that the hybrid composite had enhanced mechanical properties due to better interaction between rGO-MWCNTs and the polymer matrix. Inuwa et al. [12] investigated the effects of graphene nanoplates and multi-walled carbon nanotube hybrid nanofillers on the mechanical and thermal properties of reinforced polyethylene terephthalate. They found that the combination of the two nanofillers in the composites resulted in the overall improvement of adhesion between the fillers and the matrix, which contributed to improving the properties of the composites. Pradhan et al. [13] added multi-walled carbon nanotube–graphene (MWCNT–G) hybrids as reinforcing filler into silicone rubber (VMQ), and investigated the mechanical and thermal properties of the nanocomposites. They also found a stronger interfacial interaction between the MWCNT–G and the VMQ matrix, which contributed to improving the properties of VMQ. In other words, the interfacial adhesion significantly affected the carbon-reinforced polymeric properties [14,15]. 

In recent years, molecular dynamics simulations are used to investigate interfacial adhesion for graphene/CNT/polymer systems. Liu et al. [16], investigated the interfacial mechanical properties between hybrid graphene–CNT (GR–CNT) and a polyethylene matrix using molecular dynamics simulations. They analyzed the influences of the alignment, length and radius as well as the type of CNTs on interfacial adhesion, and showed that hybrid GR–CNT can effectively enhance interfacial mechanical properties. Zhang et al. [17] studied the load transfer of a graphene/CNT/polyethylene hybrid nanocomposite by molecular dynamics simulations. The CNT radius and the distance between a CNT and graphene had little effect on the mechanical properties of composites. However, the location of the CNT had a certain effect on the shape of the failure region during the tensile process in composite materials. 

Although it has been found that the interfacial adhesion between carbon nanoparticles plays an important role in the synergistic effect of a CNT and graphene, the mechanisms have not been well understood. In the present work, we focused on the interfacial adhesion of graphene with the matrix in epoxy/graphene/CNT composites. The pullout process of graphene from the composite model was conducted to investigate the effects of CNTs on interfacial adhesion using molecular dynamics simulations. The effects of the CNT radius, the distance between a CNT and graphene, the CNT axial direction, and the number of CNT walls were analyzed.

## 2. Calculation Models and Methods

### 2.1. Simulation Models

Diglycidyl ether of bisphenol A (DGEBA) and cyclohexylamine (CHA) were chosen as the epoxy monomer and linker, respectively (Figure 1a,b). The copolymerized chain consisting of two DGEBA and two CHA monomers (DGEBA–CHA–DGEBA–CHA) is considered an epoxy molecule, and was used to construct the epoxy matrix with a density of 1.12 g/cm^3^. Thus, the finial copolymerization degree of the polymer matrix was 75% (i.e., 3 of 4 epoxy groups were opened and chemically bonded to CHA). A single graphene sheet with the size of 43.46 × 43.79 Å^2^ and a CNT with a length of 42.54 Å were built into the simulation cell—the number of CNT walls and the CNT radii varied. Hydrogen atoms were added to the edges of the CNT and the graphene to saturate the dangling bonds. The whole simulation cell (Figure 1c) size was 46.00 × 46.00 × 46.00 Å^3^. In this work, we studied the effect of the CNT radius, the distance between the CNT and the graphene sheet, the CNT axial direction and the number of CNT walls on the interfacial adhesion between the graphene sheet and the epoxy matrix. Different values were used for each parameter mentioned above and are listed in Table 1. When we studied one parameter, the other three parameters were unchanged. The constants for the CNT radius, the distance between the CNT and the graphene sheet, the CNT axial direction and the number of CNT walls were 5.43 Å, 3.4 Å, 90°, and 1, respectively. Each initial structure was built and simulated using the Materials Studio software package (Accelrys, Inc. http://accelrys.com/products/materials-studio/ (date accessed: January 12, 2011)).

### 2.2. Dynamic Simulations

In this work, the Condensed-Phase Optimized Molecular Potentials for Atomistic Simulation Studies (COMPASS) force field was used [18]. It is a commonly used, well-calibrated hydrocarbon force field [19,20,21,22]. The Ewald method and the atom-based method were used for the Coulomb interactions and the van der Waals interactions, respectively.

Two dynamic simulations were performed to obtain the equilibrium structures of the epoxy/graphene/CNT composites. First, to relax the epoxy matrix, a molecular dynamics simulation was adopted for 2 ns at 400 K with all the carbon atoms of the CNT and graphene being fixed. This meant that all the atoms, including the CNT and graphene, were able to move freely. Another molecular dynamics simulation was adopted for 4 ns at 300 K to relax the whole model. Each molecular dynamics simulation was performed with constant volume, temperature, and number of particles, which is also known as the NVT ensemble. The same procedure was used for all composite models. 

Once the equilibrium stage was completed, the model was subjected to the pullout simulation, as in Reference [14], of the graphene sheet. First, an 80 Å vacuum layer was added in the z direction to avoid the effect of periodic boundary conditions, while the x and y directions stayed periodic. Next, the right end of the graphene and the left end of the CNT/epoxy matrix were fixed (Figure 2). Then, fixed atoms at the end of the graphene sheet were moved along the z direction to simulate the graphene pullout process. The displacement was set to 5 Å for each pullout step. The total displacement of the graphene was 50 Å. After each pullout step, a dynamics simulation of 50 ps was conducted at 300 K to equilibrate the structure. 

### 2.3. Calculation of Energies

In this work, two kinds of energies, the interaction energy and the interfacial adhesion energy, were calculated to analyze the interfacial adhesion of graphene and the polymer matrix (including CNT and epoxy). The interaction energy between the graphene and the matrix was calculated by Equation (1) below,
(1)$${\Delta E=E}_{\mathrm{total}}-(E_{\mathrm{graphene}}+E_{\text{CNT/epoxy}})$$
where *E*_total_ refers to the total energy of the entire system (the graphene sheets, CNT and epoxy). *E*_graphene_ refers to the energy of the graphene sheet. *E*_CNT/epoxy_ refers to the energy of the CNT and all the epoxy molecules. Based on this definition, a negative interaction energy means there is an exothermic bonding between the graphene and the polymer matrix. 

Equation (2) was used to calculate the interfacial adhesion energy (γ) of graphene,
(2)$$\gamma =\frac{\Delta E}{2A}$$
where Δ*E* is the interaction energy between the graphene and the polymer matrix, calculated by Equation (1). *A* is the contact area between the graphene and the polymer matrix. The interfacial adhesion energy indicates the interaction energy per unit area.

## 3. Results

### 3.1. Effect of the CNT Radius

In this work, three composite models with different CNT radii (2.71, 4.07, and 5.43 Å) were built to study the effect of CNT radius on the interfacial adhesion between graphene and polymer matrix. The equilibrium structures of the three models after the equilibrium molecular dynamics simulations are shown in Figure 3. In general, the equilibrium structures are similar to their initial structures in all three models. The CNT and graphene almost remain in their original position, and stay unchanged.

To analyze the pullout process of graphene from the composite model, the interaction energies at every pullout step were calculated (Figure 4). The interaction energies between graphene and the polymer matrix before the pullout process (at 0 Å displacement) were −832.1, −870.7, and −922.4 kcal/mol for models with a CNT radius of 2.71, 4.07, and 5.43 Å, respectively. A negative interaction energy indicated that there was absorptive interaction between the graphene and the polymer matrix. We found that, the longer the CNT radius, the contact area between the CNT and the graphene surface increased. The π–π interaction between the graphene and the CNT (the match in symmetries of phenyl groups) was much larger than the interaction between graphene and epoxy. Thus, the graphene/polymer composite was enhanced by the CNT with a longer radius. We suppose that a CNT with a longer radius than our considered values will have a greater impact on the enhancement of interfacial adhesion. The pure graphene/polymer composite model without a CNT (Figure 5) was also simulated as a comparison. The interaction energy was −793.8 kcal/mol for a pure graphene/epoxy composite model, which was lower than that of any CNT/graphene/polymer model. It suggests that the addition of a CNT strengthens the interaction between graphene and the matrix.

For all three models, the absolute values of interaction energies decreased with the increase of the graphene displacement. During the pullout process, the contact area between the graphene surface and the matrix became smaller, leading to the decrease in interaction energy. The interaction energies tended to be zero when the graphene displacement was 50 Å, in each model, since the graphene sheet was totally pulled out from the polymer matrix.

The interfacial adhesion energies of graphene during the pullout process are shown in Figure 6. For all three models, the absolute values of interfacial adhesion energies increased first, and reached their peak value when the displacement was at 15 or 20 Å. Then they decreased with the pullout of the graphene sheet. At the beginning of the graphene pullout process, the contact area between the CNT and the graphene surface accounted for a larger proportion of the total contact area between the graphene and the matrix. Therefore, the interfacial adhesion energies became higher than in the initial models, since the interaction between the graphene and the CNT was much larger than that of graphene and epoxy. In the later stages of the graphene pullout process, the graphene passed through the CNT. The contact area between the graphene and the CNT decreased, leading to a reduction in the interfacial adhesion energies for all three models.

### 3.2. Effect of the Distance between the CNT and Graphene

Three initial distances between the CNT (8, 8) and graphene (3.4, 6.8, 10.2 Å) were chosen to investigate the effect of the CNT location on interfacial adhesion. The radius of the CNT was constant at 5.43 Å in all three models. Their equilibrium structures are shown in Figure 7. When the initial distance between the graphene and the CNT was 3.4 Å, they were directly in contact with each other, with no epoxy between them. As the CNT moved farther away from the graphene, epoxy molecules moved into the space between them. The relative concentration of the composite showed a more detailed adsorption behavior of the epoxy molecules. As can be seen in Figure 7, the two deep valleys close to the graphene peak indicate the interfacial vdW-excluded regions between the graphene and the epoxy matrix. The first peak close to the left of the graphene indicates an intensification of the epoxy density (i.e., adsorption), which plays an important role in enhancing the mechanical properties of the epoxy matrix. The number of peaks between the CNT and to the right of the graphene gradually increases with increasing *d*, indicating migration and adsorption of the epoxy molecules between them.

The interaction energies and the interfacial adhesion energies of the equilibrium structures at every step of the pullout process were calculated (Figure 8) to analyze the interfacial adhesion between the graphene and the matrix. A longer distance between the graphene and the CNT gave a smaller interaction energy and a smaller interfacial adhesion energy. When the CNT was further away from the graphene, the interaction between them was weaker, leading to a decrease in the interfacial adhesion between the graphene and the matrix.

### 3.3. Effect of the Angle between the CNT Axial Direction and Pullout Direction

The angle between the CNT (10, 10) axial direction and pullout direction were set to 0°, 30°, 60°, and 90° in models A, B, C, and D, respectively. The equilibrium structures of models A to D are shown in Figure 9. The interaction energies and the interfacial adhesion energies were also calculated (Figure 10).

For all four models, the interaction energy became lower during the pullout process due to a decrease in the contact area between the graphene and the matrix. Before pulling the graphene sheet out (at 0 Å displacement), the interaction energies in the four models were similar, around 975 kcal/mol. This indicates that the angle between the CNT axial direction and the pullout direction has little effect on the interfacial adhesion of graphene and the matrix in the equilibrium structures. In the early stages of the graphene pullout process (0 Å < displacement < 20 Å), the interaction energy increased the angle between the CNT axial direction and the pullout direction. The interaction energy of model A is the lowest, and model D’s interaction energy is the highest. In model A, the proportion of interaction between the graphene and the CNT was unchanged during the pullout process, because the angle between the CNT and the graphene was 0°. However, the angle was 90° in model D. At the beginning of the pullout process, the contact area between the graphene and the CNT stayed the same, while the contact area between the graphene and the epoxy decreased with an increase in the graphene displacement. This led to a larger proportion of the interaction between the graphene and the CNT in the total interaction between the graphene and the whole matrix. The interaction between the graphene and the CNT was larger than that of the graphene and epoxy, so model D shows the largest interaction energy at the beginning of the pullout process. In the later pullout stages (25 Å < displacement < 50 Å), the angle between the CNT axial direction and the pullout direction had a completely opposite effect on the interaction energy, i.e., the larger angle gave a smaller interaction energy. The interaction in model D was weakest, and model A’s interaction energy was the largest. In the later stages of the pullout process, the graphene passed through the CNT, leading to a decrease in the contact area of the graphene with the CNT. However, in model A, the proportion of the interaction between the graphene and the CNT stayed the same. Thus, model A provided the largest interaction energy. Based on the above analysis, the angle between the CNT axial direction and the pullout direction has little effect on the interfacial adhesion of graphene in the equilibrium structure, but it plays an important role in the graphene pullout process.

Figure 10b shows the interfacial adhesion energy between the graphene and the matrix with different CNT axial directions during the pullout process. When the displacement was 0 Å, the four models had a similar interfacial adhesion energy value, indicating that the CNT axial direction does not affect interfacial adhesion. At the beginning of the pullout process, model D had the largest interfacial adhesion, since the proportion of interaction between the CNT and the graphene grew. When the displacement was larger than 25 Å, model D gave the smallest interfacial adhesion energy, because the proportion of the contact area between the graphene and the CNT decreased. The profiles of interfacial adhesion energy provided the same result as that of interaction energy.

### 3.4. Effect of a Double-Walled CNT Compared with a Single-Walled CNT

To analyze the effect of the number of CNT walls on the interfacial adhesion of graphene with the matrix, a composite model with a double-walled CNT (radius of the outer layer was 5.43 Å) was simulated (Figure 11). Its interaction energy and interfacial adhesion energy were calculated during the graphene pullout process (Figure 12). To facilitate comparison, the results of the model with a single-walled CNT (radius was 5.43 Å, see Figure 3) are shown here again.

The interaction energy and interfacial adhesion energy both suggest that the interfacial adhesion of graphene with the matrix including a double-walled CNT is larger than that of a single-walled CNT. For the single-walled CNT, only a single layer of carbon atoms interacted with the graphene sheet, while two layers of carbon atoms took effect with the double-walled CNT. Thus, the interfacial adhesion between the graphene and the matrix with a double-walled CNT was larger. In addition, the profile of the interfacial adhesion energy for the model with a double-walled CNT was more stable during the whole pullout process than that of a single-walled CNT (Figure 12b). It indicates that a double-walled CNT not only strengthens the interaction between the graphene and the matrix, but also makes the interaction more stable.

## 4. Conclusions

Molecular dynamics simulations were conducted to investigate the interfacial adhesion of graphene with a CNT/epoxy matrix based on the graphene pullout process. The effects of the CNT radius, the distance between the CNT and the graphene, the CNT axial direction, and the number of CNT walls on interfacial adhesion were analyzed. We conclude that:

1. The addition of a CNT strengthens interfacial adhesion between graphene and the polymer matrix. A larger CNT radius gives a larger area in contact with graphene, leading to a stronger interfacial adhesion of graphene with the matrix.

2. When the CNT is farther away from the graphene sheet, the interfacial adhesion of graphene with the matrix becomes weaker, due to the interaction decreasing between the CNT and graphene.

3. The CNT axial direction has little effect on the interfacial adhesion of graphene in the equilibrium structure, but it plays an important role in the graphene pullout process. At the beginning of the graphene pullout process, a larger angle between the CNT axial direction and the pullout direction gives a stronger interfacial adhesion, however, it offers a weaker interfacial adhesion at the later pullout stages.

4. The interfacial adhesion between graphene and the matrix becomes stronger when a double-walled CNT is added to the matrix compared with a single-walled CNT.

## Figures and Tables

**Figure 1 polymers-11-00121-f001:**
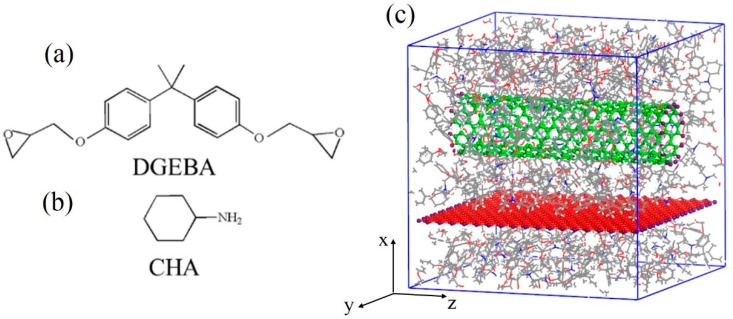
Chemical structures of DGEBA (epoxy monomer) (**a**), CHA (linker) (**b**), and view of the epoxy/graphene/CNT composite model (**c**).

**Figure 2 polymers-11-00121-f002:**
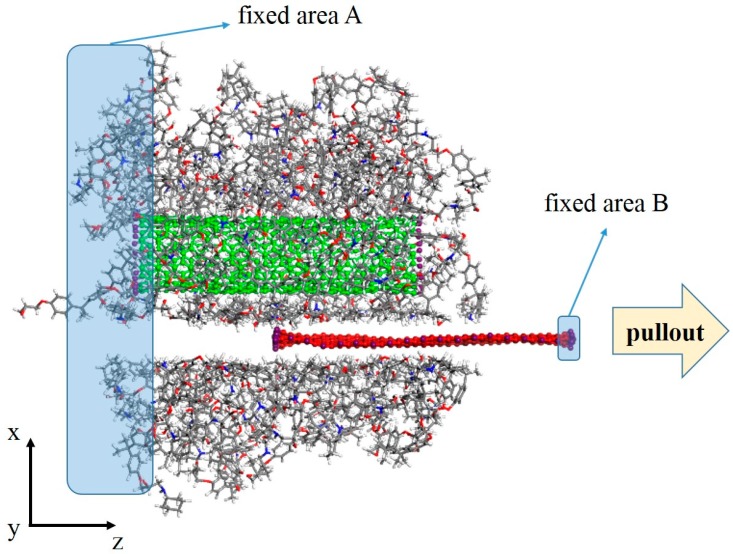
Schematic diagram of the graphene pullout process from the composite model.

**Figure 3 polymers-11-00121-f003:**
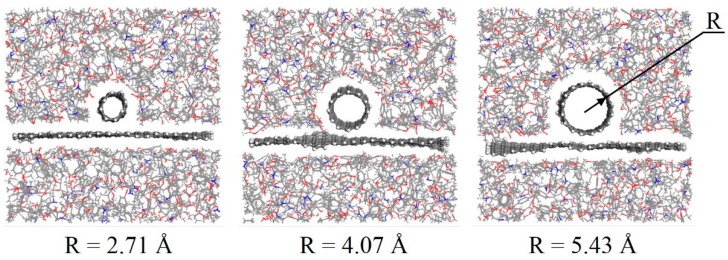
Equilibrium structures of composite models with different CNT radii.

**Figure 4 polymers-11-00121-f004:**
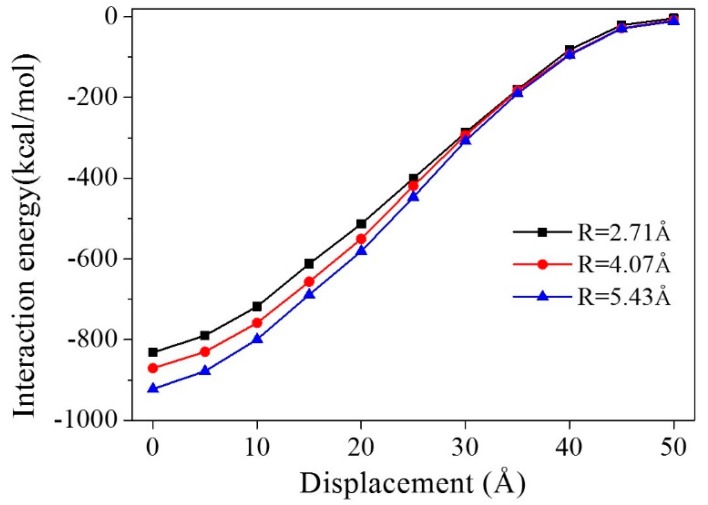
Interaction energies of graphene and a polymer matrix in composite models with different CNT radii during the pullout process.

**Figure 5 polymers-11-00121-f005:**
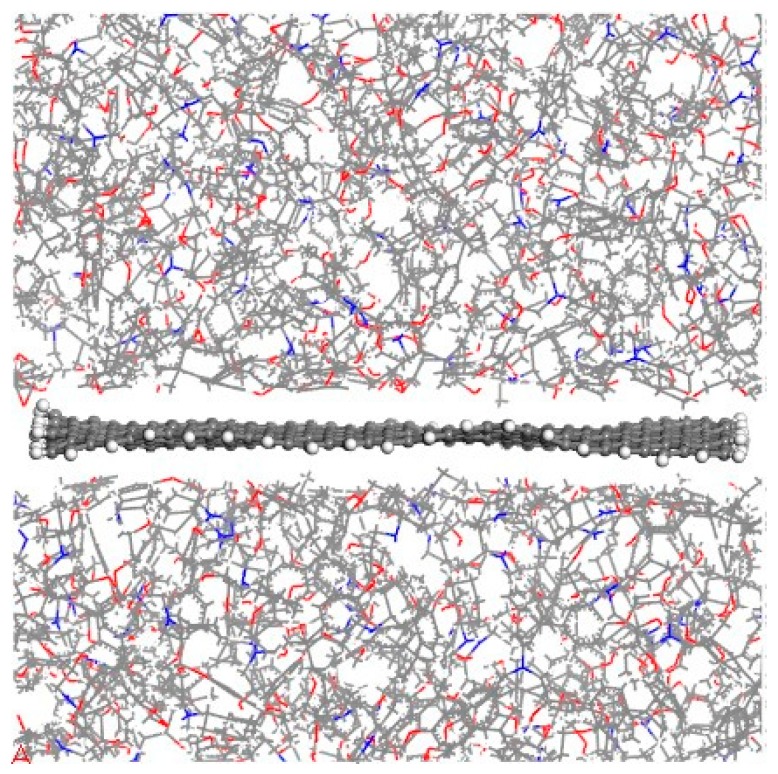
Equilibrium structure of a pure graphene/epoxy composite model.

**Figure 6 polymers-11-00121-f006:**
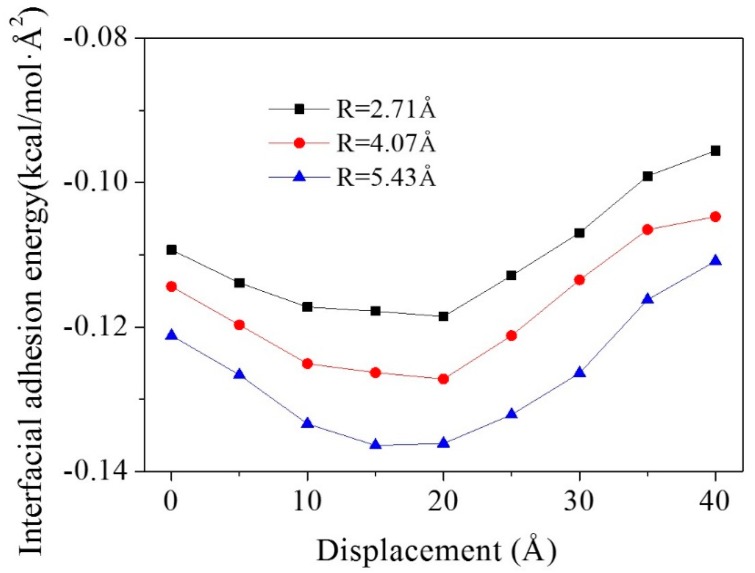
Interfacial adhesion energies for composite models with different CNT radii during the pullout process.

**Figure 7 polymers-11-00121-f007:**
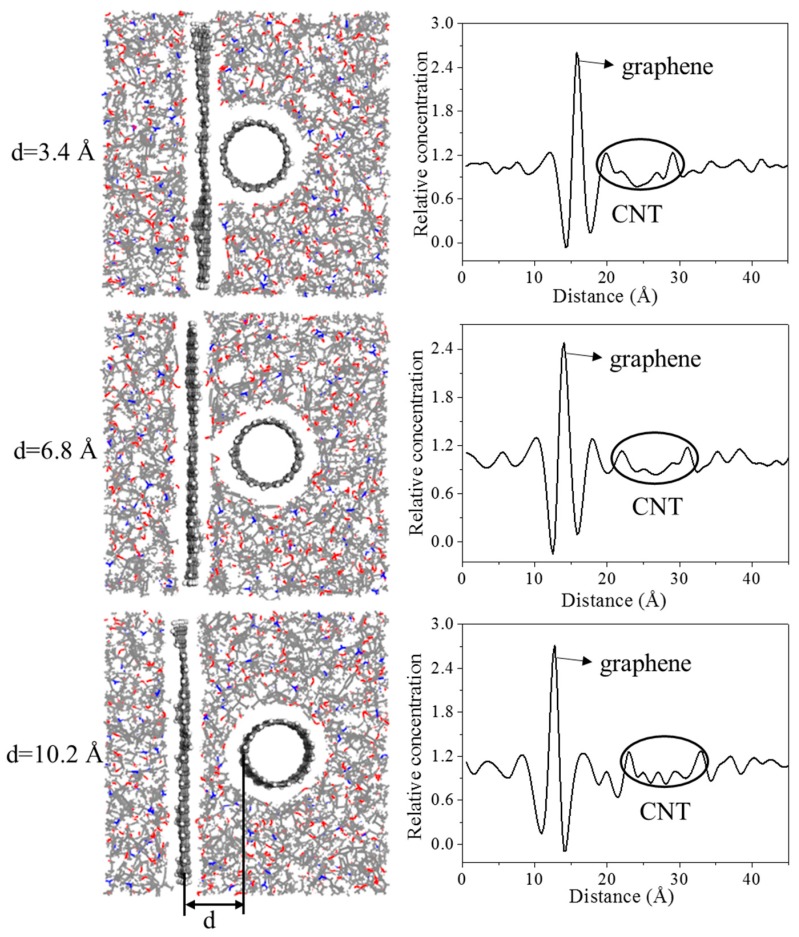
Equilibrium structures and relative concentrations of composite models with different distances between the CNT (8, 8) and graphene.

**Figure 8 polymers-11-00121-f008:**
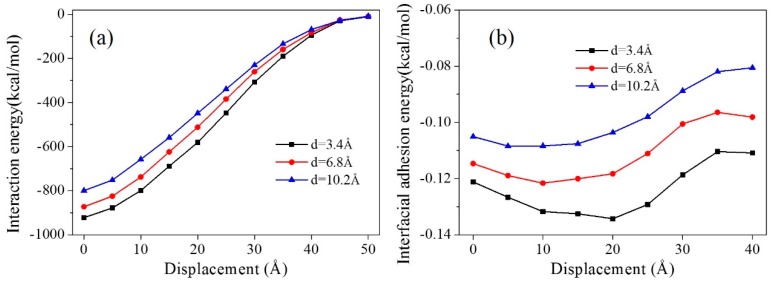
Interaction energies and interfacial adhesion energies for composite models with different distances between the CNT and graphene during the pullout process.

**Figure 9 polymers-11-00121-f009:**
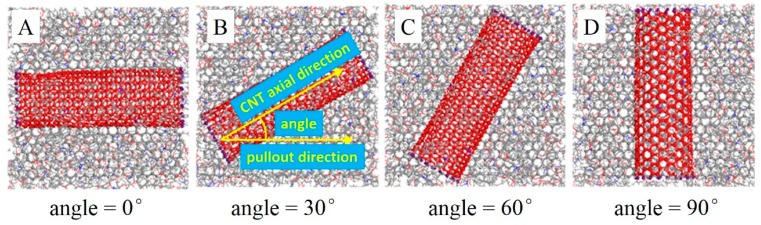
Equilibrium structures of composite models with different angles between the CNT (10, 10) axial direction and pullout direction—epoxy atoms are hidden.

**Figure 10 polymers-11-00121-f010:**
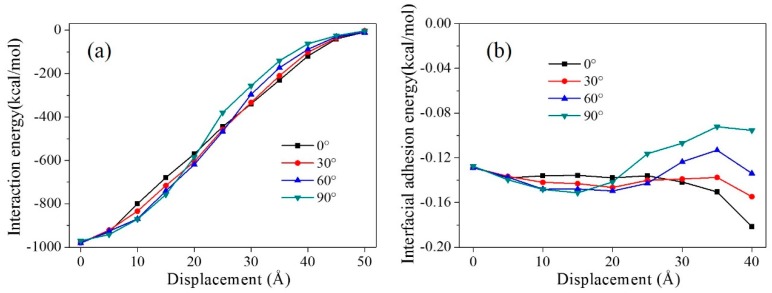
Interaction energies and interfacial adhesion energies for composite models with different angles between the CNT axial direction and pullout direction during the pullout process.

**Figure 11 polymers-11-00121-f011:**
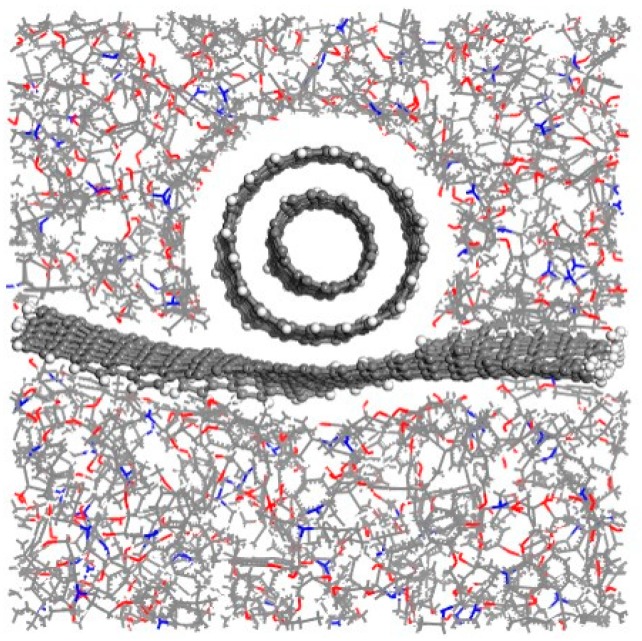
Equilibrium structures of composite models with a double-walled CNT.

**Figure 12 polymers-11-00121-f012:**
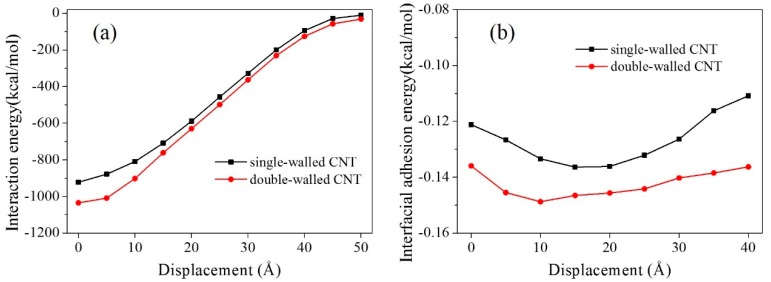
Interaction energies and interfacial adhesion energies for the composite models with a single-walled CNT and a double-walled CNT.

**Table 1 polymers-11-00121-t001:** Parameters used in different composite models.

Studied Parameters		Other Constant Parameters			
Name	Values	CNT Radius	Distance between the CNT and the Graphene Sheet	CNT Axial Direction	Number of CNT Walls
CNT Radius (Å)	2.71, 4.07, 5.43	-	3.4	90	1
Distance between the CNT and the Graphene Sheet (Å)	3.4, 6.8, 10.2	5.43	-	90	1
CNT Axial Direction (°)	0, 30, 60, 90	5.43	3.4	-	1
Number of CNT Walls (/)	1, 2	5.43	3.4	90	-

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
