# Peer review of "Effect of Carbon Nanotube Addition on the Interfacial Adhesion between Graphene and Epoxy: A Molecular Dynamics Simulation"

_polymers, 2019, doi:10.3390/polym11010121_

Round 1

Reviewer 1 Report

While the topic is of potential interest, the manuscript is lacking sufficient description of methods and analysis and hence its scientific metrit cannot be adequately judged. In addition, the main conclusions of the authors are not directly supported by the analysis.

The following comments should be addressed before consideration towards publication:

1. The authors need to describe how crosslinking of the epoxy was carried out in Methods.

2. While the authors measure a “negative interaction enegy”, the gap between the graphene/CNT and the epoxy matrix in Figure 3 suggest that the interaction is unfavorable. Clealry, there is no absorption, contradicting the authors claim that on line 147. While such scenario is expected with polar solvents, it is surprising to see it with a hydrophobic polymer. Could it be a byproduct of the simulation set-up or choice of force field?

3. The conclusion in line 152-154 is flawed as there is a greater concentration of carbon in the latter (graphene+CNT). The authors must normalize for the concentration or compare agains CNT+CNT or graphene+graphene cases.

4. Lines 191-193: What influences DE is the interaction between CNT and epoxy, not the interaction between CNT and graphene. The E_CNT/epoxy term in Eq. 1, is most influential. As d increases, it clearly becomes more negative due to increase contact area between the CNT and epoxy.

5. Why does the 0 angle in Fig. 10  not have the same shape as those shown in Fig.  8?

6. What are the radii of the DWCNTs shown in Fig. 11? it is not clear form Table 1. With which SWCNT are the results compared to in Fig. 12?

Author Response

Response to Reviewer 1

1. The authors need to describe how crosslinking of the epoxy was carried out in Methods.

Re: We use a pre-cross-linked epoxy chain DGEBA-CHA-DGEBA-CHA during model construction. The chain is linear because both DGEBA and CHA have two potential bonding sites. Therefore, a 75% crosslink degree indicates 3 of 4 epoxy groups are opened and chemically bonded to CHA. By this way, the constructed epoxy matrix is simplified and contains monodispersed epoxy chains. No further crosslinking process is conducted. To make it more understandable, we add a description on the definition of crosslink degree (line 84, revised paper). The revision is copied here:

“Thus, the finial cross-linking degree of the polymer matrix is 75% (i.e., 3 of 4 epoxy groups are opened and chemically bonded to CHA).”

2. While the authors measure a “negative interaction energy”, the gap between the graphene/CNT and the epoxy matrix in Figure 3 suggest that the interaction is unfavorable. Clearly, there is no absorption, contradicting the authors claim that on line 147. While such scenario is expected with polar solvents, it is surprising to see it with a hydrophobic polymer. Could it be a byproduct of the simulation set-up or choice of force field?

Re: Thanks for your considered analysis on this point. However, we think you may misunderstand Figure 3. The gap between the graphene/CNT and the epoxy matrix is the interfacial vDW excluded region which is reasonable for nanocomposite systems. A rough estimation of the gap width is about 1.5 Å ~ 2.0 Å. The existence of such region does not mean the interaction is unfavorable. For your references, we provide three published papers talking about this issue in nanocomposite systems.

Macromolecules 2008, 41, 1499−1511.

Macromolecules 2003, 36, 1395−1406.

ACS Appl. Mater. Interfaces 2016, 8, 11, 7499-7508.

3. The conclusion in line 152-154 is flawed as there is a greater concentration of carbon in the latter (graphene+CNT). The authors must normalize for the concentration or compare against CNT+CNT or graphene+graphene cases.

Re: The interaction energy in this paper is between the graphene sheet and the matrix (epoxy or epoxy+CNT), not between the graphene (or graphene+CNT) and epoxy. So, we think there is no problem for the discussion in line 152-154. Here, the interfacial adhesion between graphene and matrix is improved by introducing CNT, indicating that the addition of CNT has strengthened the system.

4. Lines 191-193: What influences DE is the interaction between CNT and epoxy, not the interaction between CNT and graphene. The E_CNT/epoxy term in Eq. 1, is most influential. As d increases, it clearly becomes more negative due to increase contact area between the CNT and epoxy.

Re: Here, we are talking about the interaction between the graphene sheet and the CNT+epoxy, not the interaction between CNT and graphene. With the increase of d, the contact area between graphene and epoxy increases but the case between graphene and CNT decreases. Therefore, the interaction energy reflects the combined effect of this change. We think this gives more information than the obvious enhanced interaction between CNT and epoxy with increase of d (increased contact area).

5. Why does the 0 angle in Fig. 10 not have the same shape as those shown in Fig. 8?

Re: We apologize for the unclear description. Actually, we use CNT (10, 10) in Fig. 10, and CNT (8, 8) in Fig. 8. You may find that the two lines are similar in variation trend except for different specific values. The two different CNT types have been added to the revised paper in lines 181,188,200,204.

6. What are the radii of the DWCNTs shown in Fig. 11? it is not clear form Table 1. With which SWCNT are the results compared to in Fig. 12?

Re: In Table 1, the column of “CNT radius” indicates the consistent radius of 5.43 Å for DWCNT and SWCNT. To make it easier to understand, we add the descriptions of their radii in Section 3.4. in line 242 and line 245. The SWCNT in Fig. 3 (R = 5.43 Å) is used in the comparison here.

The revisions are copied here:

“……a double-walled CNT (radius of the outer layer is 5.43 Å) was simulated (Fig.11).”

“……with single-walled CNT (radius is 5.43 Å, see Fig. 3) are shown here again.”

Reviewer 2 Report

The manuscript by Sun et al. discusses the effect of CNT addition on the adhesion of epoxy resin at the interface of graphene. The effect of CNT curvature and its distance on the adhesion is simulated and discussed. The manuscript is written in an understandable way and the simulations done are technically correct. However, some points (as addressed bellow) need more clarifications before publication.

1-Dynamic simulation: The way the authors remove pbc along the x direction is clear. However, it is not clear from the text either the simulation box is periodic along the y and z directions or not. The authors have written that “the right end of graphene and the left end of CNT/epoxy resin matrix were fixed”. How do they do it? Is there a literature on this method (please address)? How does this method affect pbc along the z direction? I am asking this because the results on the dependence of interfacial energies on the displacement depend on the method the author remove pbc along the z direction.

2-An important factor causing energetic adhesion of DGEBA at the interface of graphene and CNT is the match in symmetries of phenylene groups with those of 6-membered rings in graphene/CNT surfaces (J. Phys. Chem. C, 2014, 118, 9841). In this respect, more favorable adhesion at the interface of flatter CNTs and dependence of interaction energies on the distance between CNT and graphene can be explained.

3-Fig. 8: the decrease in the interaction energy with increase in the CNT-graphene distance can be interpreted in terms of the formation of organized layers at the interface. At very short distances, only a single compact organized layer is formed (the thickness is around one molecular layer, which is about 0.3 nm). Increasing the distance the second (less organized layer) is formed, which obviously has a less interaction energy (compared to shorter distances at which one organized layer is formed).

4-More favorable adhesion of polymers/fluids at the interface of flatter surfaces has been addressed to in literature (Macromolecules, 2013, 46, 8680). I recommend discussing the interaction/adhesion energies in terms of molecular geometries of resin and surfaces and formation of organized layers at the interface (see comment 3).

5-A CNT with infinite diameter can be regarded as a graphene surfaces. It would be of interest to readers if the authors can show this limit in the figures or discuss it in the text.

Author Response

Response to Reviewer 2

1-Dynamic simulation: The way the authors remove pbc along the x direction is clear. However, it is not clear from the text either the simulation box is periodic along the y and z directions or not. The authors have written that “the right end of graphene and the left end of CNT/epoxy resin matrix were fixed”. How do they do it? Is there a literature on this method (please address)? How does this method affect pbc along the z direction? I am asking this because the results of interfacial energies depend on the method the author remove pbc along the z direction.

Re: Only the z direction was added with an 80 Å vacuum layer to avoid the effect of pbc, while the x and y directions stay periodic. We have added further description to line 113 in the revised paper. The selected regions are fixed meaning that atoms in these regions will not move during simulations. A constraint function in the Materials Studio software can be used to fix atoms. The added 80 Å vacuum layer in z direction blocks periodic interactions, which has the same effect as removing pbc. The fixed regions ensure that the whole graphene will be totally pulled out of the matrix at certain displacement, otherwise without constraint the graphene may move back to the matrix or rotate in the y-z plane. The stepwise pullout process imposes a gradual evolution of the contact area between graphene and matrix. Therefore, the evolution of interfacial energies can be closely related to the evolution of contact area. The Ref. [14] (Polymer. 2013, 54, 3282-3289) in our paper also performs the same pullout method in MD simulations. We have added this Ref. to line 112 where the pullout simulation is introduced.

2-An important factor causing energetic adhesion of DGEBA at the interface of graphene and CNT is the match in symmetries of phenylene groups with those of 6-membered rings in graphene/CNT surfaces (J. Phys. Chem. C, 2014, 118, 9841). In this respect, more favorable adhesion at the interface of flatter CNTs and dependence of interaction energies on the distance between CNT and graphene can be explained.

Re: Thanks for the considered analysis on this point which we totally agree. We already talked about this effect in the paper. As you may find in line 150-152, we explained that the addition of CNT with a larger radius introduces more favorable π-π interactions which will contribute to a strengthened interfacial adhesion.

3-Fig. 8: the decrease in the interaction energy with increase in the CNT-graphene distance can be interpreted in terms of the formation of organized layers at the interface. At very short distances, only a single compact organized layer is formed (the thickness is around one molecular layer, which is about 0.3 nm). Increasing the distance the second (less organized layer) is formed, which obviously has a less interaction energy (compared to shorter distances at which one organized layer is formed).

Re: Thanks for this comment. We agree that the formation of organized layers between CNT and graphene will contribute to the variation of interaction energy. As shown in Figure 7, there is no layer at d = 3.4 Å, incomplete layer at d = 6.8 Å, and a single layer at 10.2 Å. We think the morphological evolution is reasonable for large molecules like epoxy resin due to complex groups and molecular configurations. A more organized layers can be obtained in systems containing small molecules. For example, we observed organized layers in “graphene + hydrocarbon” composites in our previous studies (J Nanopart Res (2016) 18: 317; J Nanopart Res (2017) 19: 195). Here, we do not talk much about the organized layer because of the imperfect layering behavior of epoxy resin.

4-More favorable adhesion of polymers/fluids at the interface of flatter surfaces has been addressed to in literature (Macromolecules, 2013, 46, 8680). I recommend discussing the interaction/adhesion energies in terms of molecular geometries of resin and surfaces and formation of organized layers at the interface (see comment 3).

Re: We thank you for this suggestion. As discussed in the above response, we don’t think discussion from the “organized layer” is appropriate for epoxy resin materials in a confined region/distance. As one of our previous paper (ACS Appl. Mater. Interfaces 2016, 8, 11, 7499-7508) explains, epoxy resin materials may form a high density region at the interface region but there will not be obvious multiple organized layers. This is mainly attributed to the complex configuration of epoxy resins. However, we appreciate your considered analysis on our paper.

5-A CNT with infinite diameter can be regarded as a graphene surfaces. It would be of interest to readers if the authors can show this limit in the figures or discuss it in the text.

Re: Discussion on the case when CNT has further increased radius than the considered values in this work has been added to line 156. As to the infinite diameter you mentioned in the comment, this will be the case when a double layered (or overlapped) graphene is used. In that case, the interfacial cohesion will become much more complex because the aggregation behavior of the two single-layer graphene plays an important role on their interactions. In fact, we have previously studied the interfacial interactions in nanocomposite systems containing overlapped graphene in our published work (J Nanopart Res (2016) 18: 317; J Nanopart Res (2017) 19: 195). Thus, we do not think the discussion of “a CNT with infinite diameter” is appropriate in our work. However, thanks a lot for this suggestion.

Reviewer 3 Report

Dear Authors,

First of all, I wish to congratulate the authors because their work is very interesting from theoretical point of view and it contains many bibliographic references.

The novelty consists in the using of the molecular dynamics simulation to model the pullout process of graphene from the graphene / epoxy composites filled with carbon nanotube (CNT). The main objective was to theoretically analyze the effects of the some internal factors (CNT radius, distance between CNT and graphene sheet, CNT axial direction and number of CNT walls) on the interfacial adhesion between graphene and epoxy resin matrix. The interfacial adhesion was analyzed by relating the interaction energy and interfacial adhesion energy with different values of one of the internal factors, during the pullout process.

The molecular dynamics simulation is a modern method used in simulation of the composite materials and the conclusions are very well supported by the results in order to recommend the publication of this paper after some little improvements.

The research topic of the manuscript fits on the journal purpose. The figures are sufficient to understand all steps of the investigations carried out during the research.

The manuscript provides important contributions to the research concerning to the effect of carbon nanotube addition on the interfacial adhesion between graphene and epoxy resin.

In order to improve this interesting manuscript, I recommend the following little changes:

Title:

I’d like to propose a little change of the title: replacing of the “… adhesion of graphene and epoxy resin…” with “… adhesion between graphene and epoxy resin…”.

Section “Abstract”:

-Line 14: In order to clarify the abbreviation CNT, it is better to replace the text with the following: “The pullout process of graphene from the epoxy resin/graphene composite filled with carbon nanotube (CNT)…”

-Line 17: the word “wall” should be replaced with “walls”.

-Line 25: I’d like to propose a little change of the text with: “…is stronger in case when a

 double-walled CNT is added…” or  “…is stronger in case corresponding to the

 double-walled CNT than the case corresponding to single-walled wall CNT.”

Section “2.1. Simulation models”

-Why the authors chosen 46.00×46.00×46.00 Å size for the simulation model? Please add some details to justify this aspect.

-Is the length of 42.54 Å corresponding to CNT related with the size of the carbon nanotube (CNT) sold by the producers according to technical sheets of such products? Please, add some details with references to those producers.

-The same remarks regarding to the different values of CNT radius considered in this section.

-What was the reason of considering of the values 3.4, 6.8, 10.2  for the distance between CNT and graphene sheet?

Author Response

Response to Reviewer 3

1. Title:

I’d like to propose a little change of the title: replacing of the “… adhesion of graphene and epoxy resin…” with “… adhesion between graphene and epoxy resin…”.

Re: The title has been updated according to your suggestion. Thanks.

2. Section “Abstract”:

-Line 14: In order to clarify the abbreviation CNT, it is better to replace the text with the following: “The pullout process of graphene from the epoxy resin/graphene composite filled with carbon nanotube (CNT)…”

-Line 17: the word “wall” should be replaced with “walls”.

-Line 25: I’d like to propose a little change of the text with: “…is stronger in case when a double-walled CNT is added…” or “…is stronger in case corresponding to the double-walled CNT than the case corresponding to single-walled wall CNT.”

 Re: The Abstract has been updated according to the above suggestions. Thanks.

3. Section “2.1. Simulation models”

-Why the authors chosen 46.00×46.00×46.00 Å size for the simulation model? Please add some details to justify this aspect.

Re: From computational point of view, the size of a model must not be too big because this is closely related to the computing power. However, there is no strict requirement on the box size as long as the model reflects a reasonable composite system under periodic boundary conditions. Here, based on our simulation experiences and the references from literatures, we are sure that the chosen 46.00×46.00×46.00 Å size of the all-atom MD simulation models is large enough for this work.

-Is the length of 42.54 Å corresponding to CNT related with the size of the carbon nanotube (CNT) sold by the producers according to technical sheets of such products? Please, add some details with references to those producers.

Re: The selected lengths of CNTs in this work are not related to CNTs sold by producers. As you know, MD simulations are quite different from real experiments. We only use theoretical models in computer simulations. Thus, there are no related producers.

-The same remarks regarding to the different values of CNT radius considered in this section.

Re: Similar to our responses above, the selection of CNT radius is not related to producers.

-What was the reason of considering of the values 3.4, 6.8, 10.2 for the distance between CNT and graphene sheet?

Re: The value of 3.4 Å is the layer-to-layer gap in multilayered graphene, and 3.4, 6.8, and 10.2 are one, two, and three times of the gap. We think this is meaningful for studying the distance induced variation of interfacial adhesion.

Round 2

Reviewer 1 Report

While the simulated system and presented results have been clarified in the revised manuscript, the authors did not sufficiently address all of the comments in their revisions. The most important conclusions made by the authors are insufficiently supported an require further analysis.

after clarification by the authors, it is clear that an oligomeric system is being modeled, and not a crosslinked epoxy. The term “crosslinking” in this study is misleading. A more proper terminology is copolymerization. Crosslinks refer to bonds between polymer chains or segments. That is not the case here as very short linear oligomers are formed. Similarly, the authors should be careful with their use of simulation of a polymer “resin”.  

the authors have still not show sufficient evidence for adsorption, visual arguments are not sufficiently convincing. As is commonly done, including in the references generously provided by the authors, support e.g. in the form of radial distribution function (rdf) needs to be added to convincingly show adsorption. Inversion of the rdf should give an indication of the magnitude of the attractive potential. Note that rdf is a normalized function and hence the first peak should give an indication of the relative attraction for the different systems considered (point 3 in below). Alternatively, in order to claim adsorption, the authors should show intensification of the density of the polymer near the graphene or CNT.

The systems studied in this paper are clear. In order to claim an enhancement effect, however, one must have a reference state to relate to (that is independent by the      number of carbon-polymer interactions). In other words, if the energy is      normalized for (graphene+CNT) carbon content, is there still an enhancement effect?

Author Response

Point-to-Point Response to Reviewer 1:

1. after clarification by the authors, it is clear that an oligomeric system is being modeled, and not a crosslinked epoxy. The term “crosslinking” in this study is misleading. A more proper terminology is copolymerization. Crosslinks refer to bonds between polymer chains or segments. That is not the case here as very short linear oligomers are formed. Similarly, the authors should be careful with their use of simulation of a polymer “resin”. 

Re: Thank you for your careful work. We have corrected the description in terms of “crosslink” and “resin” according to your suggestion. For example, we change “cross-linking degree” to “copolymerization degree”, “cross-linked chain” to “copolymerized chain”, “cross-linker” to “linker”, as can be seen in Line 81-84 in the revised paper. We also change the term “epoxy resin” to “epoxy” in the revised paper to avoid misleading description.

2. the authors have still not show sufficient evidence for adsorption, visual arguments are not sufficiently convincing. As is commonly done, including in the references generously provided by the authors, support e.g. in the form of radial distribution function (rdf) needs to be added to convincingly show adsorption. Inversion of the rdf should give an indication of the magnitude of the attractive potential. Note that rdf is a normalized function and hence the first peak should give an indication of the relative attraction for the different systems considered (point 3 in below). Alternatively, in order to claim adsorption, the authors should show intensification of the density of the polymer near the graphene or CNT.

Re: To provide evidence for adsorption behavior, we exhibit the relative (normalized) concentration of the composite systems (Figure 7, as seen below or in the revised paper). The relative concentration plays similar role with rdf in revealing the adsorption of epoxy molecules near graphene or CNT. As indicated in Figure 7, you may find the graphene (sharp peak) and CNT (between two small peaks) regions. The two deep valleys close to graphene peak indicate the interfacial vdW excluded regions between graphene and epoxy matrix. The first peak close to left side of graphene clearly indicates intensification of the epoxy density (i.e., adsorption). The number of peaks between right side of graphene and CNT gradually increases with increasing d, indicating migration and adsorption of epoxy molecules between them. Discussions about the relative concentrations can be found in Line 188-194 in the revised paper (Figure 7). Thanks for this suggestion.

Figure 7. Equilibrium structures and relative concentrations of composite models with different distances between CNT (8, 8) and graphene.

3. The systems studied in this paper are clear. In order to claim an enhancement effect, however, one must have a reference state to relate to (that is independent by the      number of carbon-polymer interactions). In other words, if the energy is      normalized for (graphene+CNT) carbon content, is there still an enhancement effect?

Re: Thank you for your valuable advice. For this work, the reference state is the model of graphene+epoxy without CNT (Figure 5). Figure 7 shows a clear adsorption behavior of epoxy matrix near the graphene. And, for all systems in this work, we do not change the carbon content of the graphene sheet (i.e., the carbon contents of the graphene are consistent). Throughout the paper, we focus on the interaction between the graphene and epoxy+CNT (not between graphene+CNT and epoxy), and the number of carbon-polymer interactions stays constant. Based on the reference state in Figure 5, we have discussed the addition of CNT on the enhancement of the system from interaction energy (Line 145-158, Egraphene-epoxy = -793.8 kcal/mol, Egraphene-epoxy/CNT = -832.1, -870.7, and -922.4 kcal/mol for models with 2.71, 4.07, and 5.43 Å radius CNT, respectively). So, we think the current discussions in the paper are not conflict with your concern. Thanks again for your careful consideration. 

Reviewer 2 Report

All my comments are taken into account in the revised version. I recommend publication.

Author Response

Thanks for your careful work.

Round 3

Reviewer 1 Report

I appreciate you taking time to clarify the manuscript. The research and methodology are now clear, and in my opinion the work is publishable.